# Exploring the role of the UK renal social worker: The nexus between health and social care for renal patients

**Maaike L. Seekles** *, **Paula Ormandy**

School of Health and Society, University of Salford, Salford, United Kingdom

* maaike.seekles@lstmed.ac.uk

## Abstract

### Introduction

Patients living with progressive chronic kidney disease may face a variety of ongoing physical, emotional, financial and/or social challenges along the disease pathway. In most UK renal units, psychosocial support has traditionally been provided by a renal social worker. However, in recent years, the numbers of renal social workers have been declining. The specialised role is poorly understood and there is no UK research about the profession. To inform future research and guide workforce planning, this study presents the first-ever exploration of the UK renal social worker role. It aimed to map the profession's activities and reasons for involvement in patient care, as well as providing an initial evaluation of that involvement on patient wellbeing.

### Methods and analysis

This mixed-method study recruited adult renal social workers (n = 14), who completed diaries over a 4-month period, participated in a focus group, and provided secondary data (caseload data and audit files where available) to give insight into their role. The evaluation of social work involvement on patient wellbeing used a pre-post intervention design. It measured distress, anxiety and depression levels as captured by the Distress Thermometer and Emotional Thermometers. A total of 161 patients completed the pre-involvement questionnaire, and 87 (55%) returned the post-involvement questionnaire.

### Results and conclusion

The renal social worker role is creative, broad and fluid, with variations in roles linked to differences in employment and funding arrangements, configurations of the wider multidisciplinary renal team, level of standardisation of psychosocial care, availability of community services, and staff-to-patient ratios. Renal social work is different from statutory social work, and renal social workers are generally able to offer continuous rather than episodic care and support patients that would not meet strict local authority eligibility criteria. The findings showed that this support leads to significantly reduced distress and anxiety.

**Data Availability Statement:** All relevant data are within the paper and its Supporting information files.

**Funding:** This work is part of a larger Industrial Case PhD Studentship, funded by Kidney Care UK

(KCUK, https://www.kidneycareuk.org/; no grant number available) which covered the salary of MS. KCUK identified the focus of the study and therefore influenced the study design, but had no involvement in the data collection, analysis, decision to publish, or writing of this manuscript.

**Competing interests:** The authors declare no competing interests.

## Introduction

Chronic kidney disease (CKD) is a worldwide public health problem, with increasing incidence and prevalence, high costs, and poor outcomes [1]. CKD is a gradual loss of kidney function, which is usually asymptomatic in earlier stages of the disease and progresses into end-stage renal disease at an advanced stage. At this stage, renal replacement therapy—dialysis or transplantation—becomes necessary to maintain life [2, 3]. CKD in earlier stages is usually managed by the general practitioner, and patients with advanced CKD are often under the care of specialised renal teams.

Life with progressive CKD provides many ongoing physical, emotional, financial and/or social challenges. For a considerable number of patients, these challenges result in psychosocial problems [4–7]. The bi-directional relationship between psychosocial issues and health outcomes has now been well established [8]. The successful integration of services for health, mental health, and social care for long-term conditions (including CKD) is high on the policy agenda [9, 10]. In most UK renal units, such integration is sought through some form of co-location and joint working of specialised psychosocial and medical staff.

At a global level, there is limited information about the organisation and delivery of psychosocial services to patients within renal units. Research and guidelines from the US, Canada, and the Netherlands suggests that in those countries, renal social workers (RSWs) are the main providers of dedicated renal psychosocial support [11–13]. Traditionally, this was the same in the UK, but a recent workforce audit found that whilst the numbers of psychologists and counsellors within the renal team had increased, the numbers of specialised social workers had declined [14]. With the RSW profession seemingly under threat, a growing number of renal patients are now reliant on local authority (LA) social work services, which have tightened eligibility criteria for their support in recent years [15].

The role of the RSW role is poorly understood, not only within the renal unit, but also within local authorities (LAs) [16]. UK literature about the role is limited to a blog paper describing 'a day in the life of the renal social worker' [17]. The current study aimed to fill this gap by undertaking an in-depth exploration of the UK RSW role. It mapped RSW activities and reasons for involvement in adult patient care and sought to offer insight into RSW's contribution to patient wellbeing. This investigation formed part of a larger, nationwide study that aimed to understand how renal psychosocial services are delivered in the UK [16].

## Methods

This mixed-methods study can be divided into two parts. The first part involved an exploration of the adult RSW role; the second part consisted of an uncontrolled, pre-post evaluation of RSW involvement. Ethical approval for this study was obtained from the University of Salford (Reference: HSR1617-155) and the NHS Research Ethics Service and Health Research Authority (Reference: 17/WS/0185). In addition, all Research and Development offices from the participating Trusts confirmed their capability and capacity to host the research.

### Participants and recruitment

RSWs were recruited to act as participants as well as data collectors. A recent renal psychosocial workforce mapping [14] had identified 58 adult RSWs in the UK, who were all eligible to participate. An invitation email was sent via the RSWs' professional network, the British Association of Social Workers Renal Special Interest Group. Those who expressed an interest received an information sheet and were given at least 48 hours to digest the study information. Written informed consent was obtained from 14 RSWs: seven from England, four from Wales

and three from Scotland. To protect their identity, no further demographic information will be provided and their names have been replaced by pseudonyms.

The impact evaluation aspect of the study involved renal patients who accessed services of participating RSWs as participants. All 14 RSWs were asked to recruit a maximum of 20 patients on a consecutive referral basis over a period of 4 months. Newly referred renal patients, over 18 years old and with capacity to provide consent were invited to take part. All patients were sent or given an information sheet by the RSW, who also completed the process of obtaining informed consent. RSWs provided the patient with the pre-intervention questionnaire before or during the first appointment. The questionnaire was completed either with or without the RSW. At the end of RSW involvement, the RSW gave the participant a post-intervention questionnaire and a pre-paid return envelope. Patients sent the questionnaire directly to the research team, without the RSW being able to see the answers. To increase response rates, all participants who did not complete the post-intervention questionnaire were sent one reminder and/or re-sent the questionnaire one time. Data collection started in February 2018 and the recruitment phase closed in December 2018. The last post-involvement questionnaire was distributed in June 2019. In total, 161 renal patients were recruited into the study, who all completed the pre-involvement questionnaire. The majority of participants was between 51–60 years old, with over 70% below 60 years of age. The sample was predominantly white, mainly dialysis patients, of whom the majority had been on dialysis for less than 3 years. The majority lived together with a partner or family and were unable to work, whilst 18% of respondents was still in employment.

## Methods and data collection tools

The data that supports the findings in this paper were concurrently collected and analysed, to form an iterative interaction between what is known and what knowledge is further required. The following methods for data collection were used:

**Diaries and participant master list.**  Over a 4-month period, RSWs individually completed daily electronic diaries in Microsoft Excel. Activities were selected from a pre-determined list and the time spent (in minutes) was noted. This list was co-developed with RSWs during preparatory discussions. RSWs also had the opportunity to leave comments alongside their entries to provide more information about which issue they were trying to address and how. RSWs were also asked to keep a participant master list as a record for participant recruitment, in which they were asked to list the main reasons for their involvement.

**Focus group.**  Eight RSWs participated in a focus group about their role and activities. The discussion focused on topics related to processes of service delivery, whether these were meeting patient needs and what best practices within service delivery would be; the RSWs place within the wider multidisciplinary team; differences between RSW and LA social work; changes to the role in recent years; and views of the future of the role. The focus group took place at the University of Salford in July 2018.

**Questionnaires.**  A pre- and post RSW involvement questionnaire was designed, which included the US National Comprehensive Cancer Network's Distress Thermometer (DT) [18] as the primary measure of distress. Initially developed by Roth et al. to screen for distress in cancer patients, it has been validated for use in the UK renal population [19]. It is a simple one-item screening tool, designed to be part of health professionals' daily practice, which asks patients to rate their distress on a scale from zero (nothing) to ten (extreme). Studies using the DT in renal care are limited, but a meta-analysis of studies in oncology patients found a good balance between pooled sensitivity (0.81, 95% Confidence Interval (CI): 0.79–0.82) and pooled specificity (0.72, 95% CI: 0.71–0.72) at the cut-off score of 4 when comparing the DT to other

diagnostic tools, such as the Hospital Anxiety and Depression Scale and Beck's Depression Inventory [20]. In addition to the DT, emotional thermometers for anxiety and depression [21] were added to the questionnaire, to allow for more detailed identification of emotional difficulties. Furthermore, the Problem Checklist asked patients to specify any issues they faced in the last week and a final question asked participants to describe which issues they wanted to receive support for. Tick box data was collected on patient demographics: age, gender, employment situation, living situation, time on dialysis and ethnicity.

**Secondary data.** Where available, RSWs provided data on their yearly referral numbers, active caseloads and any audit documents.

### Data analysis

Firstly, to explore the scope of RSW, data from all sources (qualitative comments in diaries, questionnaires, participant master list, RSW case load information and focus group discussion) were used. Any data on patient issues or actors involved was triangulated, coded and thematically analysed, to identify the variety of issues that RSWs concern themselves with. Themes represented categories of patient issues, and were developed through an iterative, inductive process of comparison of different issues that were coded under the different categories. The final coding framework, consisting of 8 domains, is presented as a figure representing the RSW role in the results section.

To identify RSW activities in response to these issues, all individual diary files were cleaned. Excel was used to calculate totals (in minutes) and percentages of time spent by each RSW for each activity category. This data was transferred into STATA, which was used to create a box plot showing the minimum, maximum and interquartile ranges, in addition to median values of percentage of time spent on each activity.

The focus group recording was transcribed verbatim and data was imported into Nvivo software for analysis. Initial findings of the need for a specialised social work role being questioned and variations in RSW activities guided an hybrid coding process. Codes were created to reflect key differences between RSW and generic social work and how these relate to the needs of the patients, in addition to reasons for the variation in activities. Member checking during informal follow-up calls with RSWs (n = 5) ensured reliability and validity of findings.

Finally, data from the pre-and post- involvement questionnaires were entered into STATA for analysis. Distress, anxiety and depression were examined through binary variables, with scores of $\geq 4$ denoting 'caseness' [18]. Eight patients did not provide a DT score on either the pre-intervention questionnaire or the post-intervention questionnaire, or both. Five patients did not complete questions about their levels of anxiety and depression in the pre- or post-intervention questionnaires. Missing data was excluded on a case-by-case basis. Descriptive statistics, including frequency tables and chi-square tests, were used to explore whether there were differences in pre-intervention characteristics and emotional issues in those who did and did not respond to the post-intervention questionnaire. Three separate, exact McNemar tests were run to determine whether there was a difference between prevalence of distress, anxiety and depression before and after RSW involvement. The significance level was set for 5%.

## Results

### Exploring the scope of RSW

Findings showed that RSW responsibilities and involvement relate to a large number of different patient issues that can manifest. RSWs would either try to solve such issues directly, or through advocating, referring or liaising with other entities.

RSW involvement was categorised relating to issues within eight prominent domains:

- activities of daily living, including issues around personal care and equipment.

- finances and benefits, mainly relating to welfare advice and employment issues.

- housing, including issues with living situations and appropriateness of housing for home haemodialysis.

- treatment, mainly relating to supporting patients along their journey across the whole renal pathway.

- caregiver support, sometimes also after the patient has passed.

- mental health, including ongoing support for patients with low mood and liaison for patients with severe mental health issues.

- social life, supporting relationships and social activities.

- legal issues, safeguarding and other issues that requires RSWs specific knowledge of the law.

Examples of issues within these domains are listed in the coloured boxes in Fig 1. An overview (although non-exclusive) of the large variety of possible agents and organisations that the RSW could interact with as part of their involvement is also mapped onto Fig 1 in the white boxes.

Focus group data revealed that RSWs do not have set interventions that can be applied to patients:

*'We do whatever we need to do to help patients cope with their renal disease. It sounds so simple, doesn't it? But everyone is so different, and everyone's experience of renal failure is so different'.*

*–Jennifer, RSW.*

As such, the RSW role is broad and seems to entail:

*'. . .anything that improves the patients' quality of life'*

*–Jennifer, RSW.*

The RSWs explained that patients rarely need support for a single issue, but that often, a complex web of issues emerges. As such, RSW involvement can be much more intensive than initially expected upon referral, and activities and focus of involvement can shift as time goes by.

*'You never know what might come up during an assessment. Last week, I went to a patient to complete a benefits form, but we ended up discussing his feelings and options for withdrawal from dialysis instead.'*

*–Megan, RSW.*

*'So someone comes in and says: "I've got dialysis, but I can't afford to come in, can I get free transport?" "Why can't you afford to come in?" You find out that they might not be on the benefits they are entitled to, they have used up all their savings, they are on a tiny, tiny income. They are not eligible for benefits because they have got no entitlement to public funds or they are using the money for some other issue and they can't live within the situation that they've got. That then leads to depression, anxiety, their house is going to be taken away from them,*

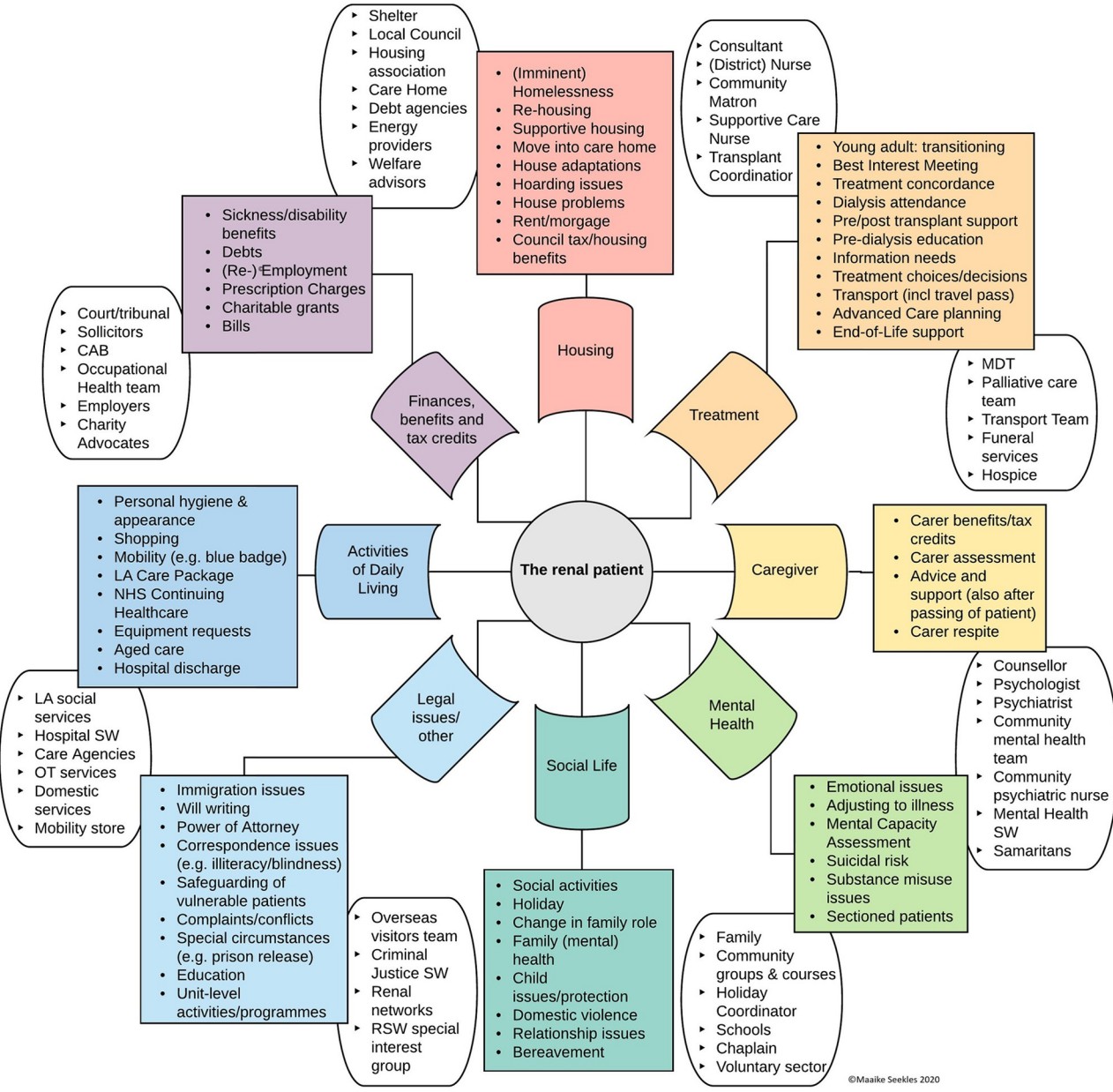

**Fig 1. The scope of renal social work: Patient issues and interacting agents.**

*the gas bill can't be paid, the electric bill can't be paid and it's a spiral of all these practical issues that can be unpicked and solved one by one that leads them to poor adherence and poor dialysis treatment for example.'*

*–Debra, RSW.*

### RSW activities

Data from diaries provided insight into the key activities that RSW involvement entails. Fig 2 provides insight into patterns and spread in terms of time spent on key activities of RSWs.

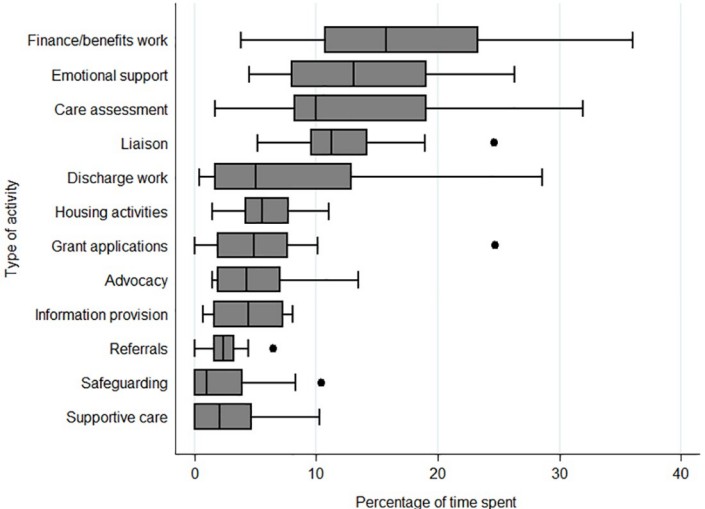

**Fig 2. Main RSW activities, based on time spent per activity.**

Activities that were not directly patient-related, such as attending meetings, travel time and administrative tasks, are not reflected in this figure.

Fig 2 shows that large variations exist between how much time each RSW spent on varying activities, or whether the RSW even did certain activities, such as supportive care or hospital discharge work. For example, time spent on financial issues varied from 3.8% to 36.2%; and whilst one RSW hardly spent any time (0.4%) on discharge activities, another spent 28.6% of their time on this. It should be noted that benefits work is not part of the social work qualification. The RSWs emphasised that they were not benefits experts and that they would '*gladly refer that on' (Layla, RSW)*. However, if there were no other services available to refer onto, such as hospital benefits advisors or an adequate Citizens' Advice Bureau, the RSWs felt compelled to support patients with their benefits questions.

## Focus group findings

**Possible reasons for variations in RSW activities.** Analysis of focus group data identified three possible reasons for the variations in RSW activities, namely: differences in funding and employment arrangements; differences in routine RSW involvement and availability of other services; and differences in patient-to-staff ratios:

As also stated in the renal psychosocial workforce report [22], RSWs are predominantly funded and/or employed through the LA or hospital Trust, with a very small proportion funded through charities. Generally, only those employed by LA have the authority to put in place care packages or undertake statutory assessments for other services that are provided through the LA. Instead, an important role of non-LA RSWs is advocating for patients to ensure that they receive appropriate and timely LA support:

> '*Advocacy is a big part of our specialist role. We have to phone, push the case, argue the case. If there is a need, often they'll prioritise ringing us back. Or for an occupational therapy assessment, if you don't say the right thing, they can sit on the waiting list for ages and ages.*'
>
> *–Carmen, RSW.*

*'I think that is the point, they do wait for ages. The only way for a RSW without that LA juris-diction to help is to phone up and say this person is high needs; they look healthy, but they are not because of this, this and this. That is really where a lot of our role comes in, advocating for these patients.'*

*–Maria, RSW.*

The RSWs also explained they were not all funded to cover the same treatment modalities. Some, for example, were only funded to cover the dialysis population and not to get involved in transplant care.

Although important to explore, a discussion around preferable funding/employment arrangements is outside the scope of this paper. Details of advantages and disadvantages to being either NHS or LA employed, related to supervision, management, recognition, and access to reporting systems can be found elsewhere [16].

Activities of RSWs were further found to be dependent on whether care processes in the units stipulated that RSWs would be routinely involved along certain parts of the renal path-way. For example, in some units, all new haemodialysis patients were referred to a RSW for an assessment. Yet, in other units, tasks like pre-dialysis assessments and supportive care support would be solely provided by dedicated nurses, with limited referrals to social work. The pres-ence or absence of other social care related professionals and organisations in the hospital or community further influenced the role. For example, some RSWs stated they were able to refer patients to a hospital benefits worker or a local disability advice organisation, whereas others were not.

Lastly, an important factor to consider when exploring the differences in roles is the large variation in RSW-to-patient ratios, as also reported in the workforce audit [14]. Data on yearly caseloads (provided by 12 RSWs) showed large differences between the number of patients that had been referred to the RSW in one year, from 72 to 520 patients per 1 Full-Time Equiva-lent. The RSWs explained that they would typically have some form of contact with all refer-rals. Logically, this large discrepancy means that some RSWs have much more time to spend on an intervention than others. The reach and scope of the role was found to reduce as patient ratios worsen, and poor ratios would impede pro-active ways of working:

*'Last year, I got over 150 referrals on top of my ongoing caseload. It just got too much, so I decided not to see CKD patients anymore, but only focus on haemodialysis patients. I also don't travel to the satellite units anymore, although I know it might be difficult for those peo-ple to come and see me here. I help patients more over the phone than in real time. I don't just go into the unit to chat to patients and introduce myself or advertise the service. There's just much more demand for the service than I can handle on my own.'*

*–Layla, RSW.*

The RSWs who cover large patient groups with minimal staff time stated that their resources would be mostly taken up by crisis patients, facing urgent situations such as immi-nent homelessness or visits from the bailiff. As such, the waiting times for less urgent cases would start to rise and the RSW involvement for these patients would involve more signpost-ing or offering advice by using video calls, phone or email, as opposed to home visits. Some of these ways of working could be further explored as they might increase efficiency. Yet it was also argued that visiting a patient in their home environment can be of great value for an intervention.

**RSW versus statutory social work.** RSW was found to be considerably different from statutory social work in the LA or hospital. Findings demonstrated that RSWs could be better positioned to support renal patients than statutory social workers, because they are more accessible, are able to provide holistic and more continuous care, and have specialist knowledge of the renal team and illness:

In recent years, changes in the social care system, and particularly the benefits system, have had a major impact on the RSW role. Increasingly, LA services have tightened their eligibility criteria for support and many patients are unable to access LA services unless they have reached a crisis stage. As a result, RSWs saw referrals to their services increase, as did the time they spent advocating for patients with the LA. Megan and her team had explored their referral data between 2010 and 2018 and found that referrals in the same two-month period had increased from 17 in 2010 to 71 in 2018. In 2010, their role seemed to almost exclusively consist of conducting pre-renal replacement therapy assessments and this number stayed relatively stable over the years. However, recently, referrals for benefits issues, housing issues, and requests related to immigration and homecare increased. Specifically, referrals for benefits increased from 1 in 2010 to 15 in 2018 over the same two-month period. RSWs felt that generic social services were currently not meeting patient needs:

*'The local authority only provides to those that have a need for safeguarding and care packaging. Anything beyond that is farmed out to the third sector and independent teams. Increasingly, so often, and this is my bugbear at the moment, I am referring and people are just not acting on referrals. In fact, I feel like they are tearing it up, people are just not getting responses.'*

*- Carmen, RSW.*

There was a consensus that LA eligibility assessments were not always fair and that many RSW service users would not get '*through the front door*' of social services, leaving their needs unmet. Patients were not asked the right questions to enable them to fully explain their situation and need for services:

*'The patients are being, may I say it, deliberately stopped from accessing a proper social work assessment. They [the assessors] will miss things out. People often have to wait a long time for an assessment to then hear they do not get the equipment. Then I say: "But have you told them you can't get down the stairs and about difficulties with meals etc?", then their response will be: "Well, they didn't ask".'*

*–Maria, RSW.*

Instead, the only eligibility criterium RSWs apply is whether a patient is under the care of a nephrologist. As such, RSWs are able to support patients that are not yet in the 'high need' category, for whom the LA would have no duty to assist. RSWs described their profession as an '*old school*', '*therapeutic*' version of social work, that allows them to support all patients in need.

*'Renal social work is about being creative, getting into old-school, old fashioned social work. It is about getting to know people well and getting the best for them, because you know them well.'*

*–Jennifer, RSW.*

RSWs explained that without their expertise and advocacy involvement, many patients would not be able to access community resources, or get adequate support from them. To illustrate, Maria recalled a time when she returned from long-term leave and her post had not been covered, leaving patients to try and access services on their own:

*'My answering machine basically said for this year I'm not here: go to CAB [Citizens Advice Bureau] with this problem, go to the law centre for this, or go to the council. That's it. Then I came back to find out a lot of people with high level disabilities had their Employment and Support Allowance stopped and were about to be evicted, they had their PIPs dropped to lower levels. So, I came back to a lot of problems because my patients couldn't access the community services.'*

*–Maria, RSW.*

In addition to experiencing issues related to social care, many renal patients suffer from emotional problems such as anxiety and depression. The RSWs explained that patients were not only struggling to access social services, but that community mental health services were also increasingly under pressure. Many RSWs, particularly those without renal psychology colleagues, feel like they are *'also holding that patient group [with psychological issues] afloat' (Carmen, RSW)*, by offering access to lower-level emotional support.

RSWs were not only more accessible to patients because of their inclusive eligibility criteria, but also due to being based in the hospital. Some RSWs with better staff to patient ratios were involved in doing assessments, home visits, or were able to visit the renal units on a regular basis. This then meant that patients knew of the RSW service, leading to self-referrals. This co-presence also means that the lines of communication between RSWs and MDT members are shorter than those between LA social work and MDT members.

As stated, renal patients often face a variety of simultaneous problems. In the current context of fragmented services, some patients are left to interact with many different agencies and professionals across the health and social care systems. RSWs argued that many of their service users feel too ill, overwhelmed or distressed to actively engage with these fragmented services. Others may not know which services are available, or they lack the capacity to keep an overview of their care. As such, RSWs play an important role in supporting patients by providing oversight of their care.

*'The agencies in the community are increasingly difficult to get hold of. People are living with so much distress, ill health and as a result lack motivation, strength and power, that they just give up.'*

*- Carmen, RSW.*

The RSWs acknowledged the importance of the nursing role in supporting patients with lower-level psychosocial issues. Yet, the complexity of the social services system makes this not an easy task, particularly in larger units. In these units, patients come from different areas, covered by different LAs and local organisations. This makes it difficult for nurses to keep oversight of which services are offered in the community, how they should refer to them and follow-up whether patients have been able to access the service. The RSWs argued that this is *'where they come in'*. They are able to assess all patient issues and take on a case management role, a holistic service that other providers cannot offer:

*'The nurses on the ward are not going to unpick all of the patient's problems, they are not going to have the knowledge or the time to unpick this. No one else is, if you send someone out to the homeless team they will look at the homelessness, then they have to go to CAB, then they are referred to the law centre for the next thing, then dietician, so many different places and they won't go because they've got 20 other appointments to get to. And no one, the outside agencies, they don't care enough about the whole issue to stop and unpick everything. And that is where we, RSWs, come in, because we have the knowledge and experience to do that.'*

*–Debra, RSW.*

Not only do RSWs provide holistic care, they also provide continuity of care. RSWs are able to support patients as their needs change throughout their whole renal journey, from diagnosis to end-of-life care. This was considered beneficial for the patients. RSWs argued that they could build relationships, learn about patients' supportive networks and understand what works and what does not work for them.

*'They get to know us, and that's what people don't get in Local Authority social work. They get an intervention and then it's closed and then somebody else comes along.*

*–Maria, RSW.*

This ability to build relationships was perceived to be an essential element of a good social work intervention, especially as some patients would not share personal problems before a certain level of trust was gained. The RSWs further stated that statutory social workers would close cases much earlier than they would do, especially if patients were not engaging. Since non-attendance was often a reason for the RSW to get involved in the first place, they argued that immediately giving up on a patient that does not engage would go against their objectives.

Finally, RSWs have, as the name implies, specialist knowledge of renal disease, which includes an understanding of the impact of the illness on a patient's physical and emotional wellbeing, daily schedule and wider social situation. There was a consensus that this renal expertise, in combination with being based in (proximity of) the renal unit, provides further advantages of RSW as opposed to LA social work. By acting as the mediator between medical staff and patients it allowed them to support those who experience distress due to questions or disagreements about their treatment:

*'I find a lot time that I might be mediating between medical staff and patients and I think what they hear from medical staff, how they hear it, is different to when they talk to us. We've got the link with it, but we are not clinical.'*

*–Karen, RSW.*

It also meant that the RSWs were able to benefit renal patients on a group-level. They can challenge issues that affect multiple patients, or organise peer-support activities or initiatives for several patients at the same time. Furthermore, RSWs are usually able to offer more flexibility in scheduling appointments around or during treatment times. This offers ease of access by seeing patients in the units, although some would also do home visits if preferred. As mentioned, RSW report having greater understanding and continued engagement when patients miss their appointments.

RSWs work more easily, in an integrated way, with other (renal) allied health professionals, such as psychology or occupational therapy services. This integration in the team, knowledge of the patient, and understanding the importance of treatment, is vital to be able to respond

swiftly when situations arise that could stop patients from attending their treatment. Carmen stated that since these problems were *'normally a crisis'* and *'normally chaotic'*, being at the *'front-line allows RSWs to be responsive and immediate'*. LA social services are not always able to do this, as became clear from this dialogue:

*'When I was on annual leave last year there was a carer in crisis, and they [social services] said 'we'll give you a list of care agencies', because she was self-funding'.*

–*Becky, RSW.*

*'. . .Which is against the care act'.*

–*Debra, RSW.*

*'Yes, two days later her husband was in hospital because he was so poorly, the list of care agencies hadn't even gotten to them by that time'.*

–*Becky, RSW.*

The findings showed that RSWs' specialist knowledge of renal disease was of vital importance to help patients articulate the impact that the illness had on their lives in such a way that would grant them access to the (benefit) support they were entitled to. Especially for dialysis patients, this can be complex, since their health and ability to look after themselves can fluctuate greatly throughout the day. It was said that if RSWs did not help to complete requests for support and followed up with advocacy efforts, patients were often unjustly refused.

*'As a renal social worker, you know the condition and the symptoms, so you know how to ask the right questions when filling in forms, when to explore, or get more into details. For example, someone might say 'yes' to the question if they can get dressed or can walk up the stairs, but then don't say that they will be so tired after that they have to rest for 3 hours. Non-renal staff won't know this and then patients won't get the support they should be getting.'*

–*Layla, RSW.*

*The criteria [from social services] are strict, but we can frame it in such a way, only through having the knowledge, that specialism. It is not saying that we are winging people through because we are throwing our weight about, they genuinely should meet the need. Alternatively, if they don't, then we become creative and look at third sector agencies, grants, or different ways of working, different ways of supporting people. If we didn't, I don't think things would move on for that patient or that family.'*

–*Carmen, RSW.*

## Evaluating RSW involvement using pre-and post-intervention questionnaires

This final section presents the results of an exploration of the impact of RSW involvement on self-reported distress, anxiety and depression levels. Out of 161 patients who completed a pre-involvement questionnaire, 88 (55.0%) returned a post-involvement questionnaire. There was no significant difference between respondents and non-respondents on the basis of their emotional issues and demographics as reported in the pre-involvement survey (Table 1).

**Table 1. Comparison of characteristics of post-intervention non-respondents and respondents.**

| Characteristic | Non-respondents (%) | Respondents (%) | Total (n = 161) | Comparison of proportions |
|---|---|---|---|---|
| *Emotional issues* | | | | |
| Distress | 90.0 | 94.1 | | $\chi^2 = 0.911$; p = .340 |
| Anxiety | 87.1 | 84.1 | | $\chi^2 = 0.292$; p = .589 |
| Depression | 72.5 | 66.7 | | $\chi^2 = 0.607$; p = .436 |
| *Sex* | | | | $\chi^2 = 0.952$; p = .329 |
| Male | 32 (41.0) | 46 (59.0) | 78 | |
| Female | 39 (48.8) | 41 (51.2) | 80 | |
| *Age Category* | | | | $\chi^2 = 7.95$; p = .093 |
| 18–40 | 19 (52.8) | 17 (47.2) | 36 | |
| 41–50 | 15 (45.5) | 18 (54.5) | 33 | |
| 51–60 | 19 (42.2) | 26 (57.8) | 45 | |
| 61–70 | 6 (23.1) | 20 (76.9) | 26 | |
| >70 | 11 (61.1) | 7 (38.9) | 18 | |
| *Ethnicity* | | | | $\chi^2 = 0.717$; p = .397 |
| White | 53 (43.4) | 69 (56.6) | 122 | |
| Other | 19 (51.4) | 18 (48.6) | 37 | |
| *Treatment Modality* | | | | $\chi^2 = 9.545$; p = .089 |
| Pre-dialysis | 21 (65.6) | 11 (34.4) | 32 | |
| HD | 31 (39.7) | 47 (60.3) | 78 | |
| PD | 4 (33.3) | 8 (66.7) | 12 | |
| Transplant | 11 (45.8) | 13 (54.2) | 24 | |
| Conservative care | 1 (100.0) | 0 (0.0) | 1 | |
| Carer | 2 (25.0) | 6 (75.0) | 8 | |
| *Time on Dialysis* | | | | $\chi^2 = 4.221$; p = .377 |
| <6 months | 6 (30.0) | 14 (70.0) | 20 | |
| 6 months to 3 years | 15 (42.9) | 20 (57.1) | 35 | |
| 3 to 5 years | 9 (52.9) | 8 (47.1) | 17 | |
| >5 years | 7 (35.0) | 13 (65.0) | 20 | |
| Not applicable | 32 (51.6) | 30 (48.4) | 62 | |
| *Living situation* | | | | $\chi^2 = 0.984$; p = .321 |
| Living alone | 26 (51.0) | 25 (49.0) | 51 | |
| Living together | 46 (42.6) | 62 (57.4) | 108 | |
| *Employment situation* | | | | $\chi^2 = 4.142$; p = .247 |
| Employed/in education | 18 (62.1) | 11 (37.9) | 29 | |
| Unemployed | 3 (42.9) | 4 (57.1) | 7 | |
| Unable to work | 39 (41.9) | 54 (58.1) | 93 | |
| Retired | 11 (39.3) | 17 (60.7) | 28 | |

Three separate, exact McNemar Chi-Square tests were run to determine whether there was a difference between prevalence of distress, anxiety and depression before and after RSW involvement. Even though prevalence of emotional problems remained high (Table 2), findings show significant reductions in the prevalence of distress (p = .007) and anxiety (p = .008) before and after RSW involvement, but not in the prevalence of depression (p = .252).

## Discussion

This mixed-methods study set out to undertake a first-ever exploration of the UK RSW role. It found that RSWs address a wide range of patient issues and support many patients who are

 

**Table 2. Prevalence of emotional issues before and after RSW involvement.**

| Prevalence measure | Before RSW involvement n (%) | After RSW involvement n (%) | Overall percentage change | P-value |
|---|---|---|---|---|
| Distress (n = 80) | 75 (93.8) | 64 (80.0) | -14.7% | < .01 |
| Anxiety (n = 83) | 70 (84.3) | 57 (68.7) | -18.6% | < .01 |
| Depression (n = 83) | 56 (67.5) | 52 (62.7) | -7.1% | .252 |

unable to access community social services, a growing problem which is known to also affect patients with other long-term conditions [23]. RSWs respond to patient issues through informing patients (and clinical staff) about available social care services; using their renal knowledge to help patients access the services they are entitled to; finding creative solutions to support patients for whom public services will not provide; offering continuous care; and helping patients keep an overview of the complex and fragmented aspects of their care. Indeed, the value of the RSW role to patients is in providing holistic care with a focus on advocacy, without adhering to strict eligibility criteria. This was found to significantly reduce distress and anxiety, thereby confirming findings of a recent study on the relationship between renal psychosocial staffing levels and distress [24].

The focus of RSW is on offering pro-active and preventative support, yet this is impeded by inappropriate staffing levels, in combination with high volumes of often crisis-driven referrals. In recent years, the number of RSW has reduced, whilst counselling and psychological services have increased [14]. The current results do not support such an alteration in the composition of psychosocial services. Instead, they provide a compelling case for the continuing existence of RSW *alongside* an increase in psychological support, especially at a time when LA services are increasingly reactive, fragmented and difficult to access.

Even though individual RSW roles within units may be well-established, as a whole, the RSW profession and role appears to be fluid and ill-defined. This is greatly attributable to a lack of standardised processes of psychosocial care across renal units and units not meeting recommended staffing levels. Instead of forming a routine, integrated part of renal care, RSW assessment and support is often offered at the discretion of medical staff or through self-referrals, leading to inequalities in patient access to RSW services across the country.

Workforce recommendations for UK RSWs are similar to those in other countries such as the US, Canada and the Netherlands. Practice standards in these countries stipulate a proactive way of working, offering all patients a psychosocial assessment at home as they move along the patient pathway. Offering such an assessment would bring practice in line with National Institute for Health and Care Excellence's quality standards, stipulating that all patients should receive a psychosocial evaluation before starting renal replacement therapy [3]. In addition, it would allow RSWs to work in the preventative way envisaged in the Care Act and support patients before problems escalate and affect quality of life and disease self-management. The exact benefits, and particularly cost-effectiveness, of offering every patient a psychosocial assessment before starting renal replacement therapy remain unclear and require further investigation.

RSWs appear to be stuck in a vicious circle: Evidence on the impact of their role is lacking; funding for their role is reducing while patient numbers are increasing; due to increasing pressures they become more invisible, leading to further ambiguity and questions about the necessity of the role within their team. RSWs need to be given dedicated time for research and service development, as other staff often already have. Such efforts could focus on developing and evaluating interventions that target renal outcomes (such as increased diet, medication and treatment adherence), to create a business case that makes clear the value of the RSW role

to the bottom line of the renal unit, as well as to the patient. On a unit-level, changes need to be made to ensure effective partnership working between RSWs and the rest of the renal team. Unit managers should lead the development of arrangements that foster relationships and greater understanding of roles and referral processes, as well as providing RSWs with access to adequate reporting systems, supervision, and training. One area of training that could be explored is the possibility of upskilling RSWs with a counselling qualification from an accredited University. Few RSWs already hold such a qualification. According to new guidelines in cancer care, social workers with a counselling qualification could perform psychological screening and certain psychological interventions, in addition to their social work interventions, under supervision of psychologists [25]. A similar construction in renal care could allow RSWs to become a logical first point of contact for all renal psychosocial support.

## Strengths and limitations

Funding requirements for this study left their mark on the study design. A focus on the adult RSW role was stipulated in the project plan, in addition to an emphasis on RSW capacity-building with regards to service evaluation. For this reason, RSW were actively involved in the data collection and recruitment process. For many RSWs, this was their first experience with research and their participation was a strength of the study. The success is visible, with one of the RSWs now starting their own PhD and some RSWs planning to take the Distress Thermometer and the activity diaries forward in their practice. The involvement of RSWs in the data collection process also brought with it some challenges, which highlight the need for increased social work staffing levels and dedicated time in the RSW role description for research and training to enable RSWs to further develop their service evaluation skills. Firstly, due to time restrictions and the unpredictable nature of the job, RSWs felt that they did not always have the time to introduce the study to patients, provide them with an information sheet and complete the pre-intervention questionnaire. In addition, sometimes, RSWs felt that it was inappropriate to ask a patient to participate if they were highly distressed or were dealing with sensitive concerns, such as safeguarding issues. This highlights an issue with RSWs as 'gatekeepers' to the research and could mean that due to a potential selection bias the sample is not entirely representative of the RSW population, with people dealing with more acute and complex issues not recruited. It is important to consider how this challenge might affect future attempts at service evaluations as part of RSW practice. Perhaps it can be explored whether standard practice can involve a formal intake, which includes the completion of an assessment/outcome tool before any services are offered, as is often the case in psychology services. A further limitation was that this was a prospective service evaluation, not a controlled study of RSW intervention versus standard care. Whilst the innovative use of the Distress Thermometer to re-evaluate distress over time was a strength of the study, it cannot be said with certainty that the observed improvements in distress and anxiety were directly attributed to the RSW intervention.

## Conclusion

RSWs play an important role in reducing distress and supporting psychosocial wellbeing of CKD patients. With their knowledge of both the illness and the social care system, RSWs may be essential to achieve the integration of health and social care for kidney patients that is so often recommended in policy documents. However, the role is fluid and ill-defined, and the importance of having RSW support at set points along the renal pathway is recognised in only a limited number of renal units. Now more than ever, the RSWs are required to demonstrate their unique worth. Clear standards of practice and a formalisation of the RSW role and

involvement along the renal pathway are needed to guide renal units in their future funding allocations.

## Supporting information

**S1 Data.**
(SAV)

**S2 Data.**
(SAV)

## Acknowledgments

In addition to the patients who participated in the study, the authors would like to acknowledge a core group of RSWs who have generously donated their time and shared their knowledge and experiences. We would specifically like to thank Maria Da-Silva Gane, Margaret Eyre, Andrew Barnett, Caron Jones, Dawnn Relph and Mandy Rathjen for their feedback on the manuscript and advice throughout the study. Finally, we would like to thank Paul Bristow and Kidney Care UK for making this study possible.

## Author Contributions

**Conceptualization:** Maaike L. Seekles, Paula Ormandy.

**Data curation:** Maaike L. Seekles, Paula Ormandy.

**Formal analysis:** Maaike L. Seekles.

**Funding acquisition:** Paula Ormandy.

**Investigation:** Maaike L. Seekles, Paula Ormandy.

**Methodology:** Maaike L. Seekles, Paula Ormandy.

**Project administration:** Maaike L. Seekles.

**Resources:** Maaike L. Seekles.

**Supervision:** Paula Ormandy.

**Validation:** Maaike L. Seekles.

**Visualization:** Maaike L. Seekles.

**Writing – original draft:** Maaike L. Seekles.

**Writing – review & editing:** Maaike L. Seekles, Paula Ormandy.

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
