## [Decision Letter · Decision Letter 0]

10 Jun 2022

PONE-D-22-08494Exploring the role of the UK renal social worker: the nexus between health and social care for renal patientsPLOS ONE

Dear Dr. Seekles,

Thank you for submitting your manuscript to PLOS ONE. After careful consideration, we feel that it has merit but does not fully meet PLOS ONE’s publication criteria as it currently stands. Therefore, we invite you to submit a revised version of the manuscript that addresses the points raised during the review process.

Please note that we have only been able to secure a single reviewer to assess your manuscript. We are issuing a decision on your manuscript at this point to prevent further delays in the evaluation of your manuscript. Please be aware that the editor who handles your revised manuscript might find it necessary to invite additional reviewers to assess this work once the revised manuscript is submitted. However, we will aim to proceed on the basis of this single review if possible.  Your manuscript has been assessed by an expert reviewer, whose comments are appended below. The reviewer, while broadly positive about your manuscript, has highlighted some areas where additional methodological details or restructuring would make your manuscript easier to follow and your work easier to reproduce. Please ensure you respond to each point carefully in your response to reviewers document, and modify your manuscript accordingly.

We look forward to receiving your revised manuscript.

Kind regards,

Joseph Donlan

Editorial Office

PLOS ONE

Journal Requirements:

Reviewers' comments:

Reviewer's Responses to Questions

**Comments to the Author**

1. Is the manuscript technically sound, and do the data support the conclusions?

Reviewer #1: Partly

2. Has the statistical analysis been performed appropriately and rigorously? 

Reviewer #1: I Don't Know

3. Have the authors made all data underlying the findings in their manuscript fully available?

Reviewer #1: Yes

4. Is the manuscript presented in an intelligible fashion and written in standard English?

Reviewer #1: Yes

5. Review Comments to the Author

Reviewer #1: This study is about the renal social workers that in recent years have decreased in the UK. The authors point out that the specialised role is poorly understood. The study aim was to show the content of the role of the renal social worker, reasons for involvement in the patient care and the effect of the involvement on the patients’ wellbeing. This was done with mixed methods, both a qualitative and a quantitative approach. The results showed that the role is broad and fluid with variations in the roles. The renal social workers can offer continues support and with their involvement the renal patients reported less distress and anxiety.

The manuscript is well written, and it is an interesting and important subject, not only for UK context. However I have some suggestions of improvement to make as follows below.

Participants and recruitment Page 6 line 116. Each RW recruited 20 patients – who were these RSW? The 14 included in the study? Please clarify.

Page 7 p 162 Data analysis: Please add figure and figure legends regarding the time used for different issues (from daily files)

Page 8 line p 163 Further describe the focus group and analysis in more depth stepwise.

The statistical analysis of the pre and post questionnaires could also be described in more detail.

Line 166-169 could you please explore this sections a bit more? E.g. what is secondary data? How did you get to the results in the figure?

Results: It is a bit difficult to follow the results. A clearer structure id needed in my opinion. Each method could perhaps be presented: Results from the diary, result from the focus group intervenes/discussion with quotations supporting each theme etc. On overarching theme with subthemes and some explanatory text?

Discussion: page 22 479: The discussion is long and the take home message is quite clear could the discussion therefor perhaps be shortened slightly?

Please clarify regarding the the NICE guidelines. A lot of abbreviations in the text. Perhaps the pre-RRT could be spelled out?

The abstract is very clear and describes the results and conclusion really well. The conclusion in the main text lacks the important effect with the renal social worker in the team for the renal patients’ psychological wellbeing.

6. PLOS authors have the option to publish the peer review history of their article (what does this mean?). If published, this will include your full peer review and any attached files.

Reviewer #1: No

---

## [Author Response · Author response to Decision Letter 0]

23 Jul 2022

Changes have been made to the title page to ensure that it now fits with the style requirements. 

2. In your Data Availability statement, you have not specified where the minimal data set underlying the results described in your manuscript can be found. PLOS defines a study's minimal data set as the underlying data used to reach the conclusions drawn in the manuscript and any additional data required to replicate the reported study findings in their entirety.

The minimum underlying datasets have been uploaded as two supporting files. This is now reflected in the Data Availability Statement.

The listed references have remained the same, but some further information in terms of DOI and internet links have been added to ensure references are complete.

Reviewer comments:

‘This study is about the renal social workers that in recent years have decreased in the UK. The authors point out that the specialised role is poorly understood. The study aim was to show the content of the role of the renal social worker, reasons for involvement in the patient care and the effect of the involvement on the patients’ wellbeing. This was done with mixed methods, both a qualitative and a quantitative approach. The results showed that the role is broad and fluid with variations in the roles. The renal social workers can offer continues support and with their involvement the renal patients reported less distress and anxiety.

The manuscript is well written, and it is an interesting and important subject, not only for UK context. However I have some suggestions of improvement to make as follows below.’

We would like to thank the reviewer for taking the time to thoroughly assess our manuscript. We have addressed your constructive feedback as detailed below, and feel that this has improved the quality and readability of our manuscript.

Participants and recruitment Page 6 line 116. Each RW recruited 20 patients – who were these RSW? The 14 included in the study? Please clarify.

Yes, the 14 RSWs who agreed to participate in the study were asked to recruit a maximum of new service users. This section has now been changed as follows, to clarify:

RSWs were recruited as participants as well as data collectors. A recent renal psychosocial workforce mapping (14) had identified 58 adult RSWs in the UK, who were all eligible to participate. An invitation email was sent via the RSWs’ professional network, the British Association of Social Workers Renal Special Interest Group. Those who expressed an interest received an information sheet and were given at least 48 hours to digest the study information. Written informed consent was obtained from 14 RSWs: seven from England, four from Wales and three from Scotland. To protect their identity, no further demographic information will be provided and their names have been replaced by pseudonyms. 

For the RSW evaluation aspect of the study, renal patients who accessed services of participating RSWs were recruited as participants. All 14 RSWs were asked to recruit a maximum of 20 patients on a consecutive referral basis over a period of 4 months. Newly referred renal patients, over 18 years old and with capacity to provide consent were invited to take part.

Page 7 p 162 Data analysis: Please add figure and figure legends regarding the time used for different issues (from daily files). 

Details on the creation of the figure (box plot) and figure legends (percentage of time spent per activity) have now been added as follows:

All individual diary files were cleaned and Excel was used to calculate totals (in minutes) and percentages of time spent by each RSW for each activity category. Percentage data was transferred into STATA, which was used to create a box plot showing the minimum, maximum and interquartile ranges, in addition to median values of percentage of time spent on each activity.

Page 8 line p 163 Further describe the focus group and analysis in more depth stepwise.

Further information was added about the discussion topics of the focus group as follows:

Focus group: Eight RSWs participated in a focus group about their role and activities. The discussion focused on topics related to processes of service delivery, whether these were meeting patient needs and what those would look like in an ideal world; the RSWs place within the wider multidisciplinary team; differences between RSW and LA social work; and changes to the role in recent years and views of the future of the role.

In addition, the analysis process has been described in further detail:

The focus group recording was transcribed verbatim and data was imported into Nvivo software for analysis. Initial findings that the need for a specialised role was being questioned and that there were variations in RSW activities guided an inductive coding process. Codes were created to reflect key differences between RSW and generic social work and how these relate to the needs of the patients, in addition to reasons for the variation in activities. Member checking during informal follow-up calls with RSWs (n=5) ensured reliability and validity of findings.

The statistical analysis of the pre and post questionnaires could also be described in more detail.

This has now been described in more detail as follows:

Finally, data from the pre-and post- involvement questionnaires were entered into STATA for analysis. Distress, anxiety and depression were examined through binary variables, with scores of ≥4 denoting ‘caseness’ (18). Eight patients did not provide a DT score on either the pre-intervention questionnaire or the post-intervention questionnaire, or both. Five patients did not complete the questions about their levels of anxiety and depression in the pre- or post-intervention questionnaires. Missing data has been excluded on a case-by-case basis. Descriptive statistics, including frequency tables and chi-square tests, were used to explore whether there were differences in pre-intervention characteristics and emotional issues in those who did and did not respond to the post-intervention questionnaire. Three separate, exact McNemar tests were run to determine whether there was a difference between prevalence of distress, anxiety and depression before and after RSW involvement. The significance level was set for 5%.

Line 166-169 could you please explore this sections a bit more? E.g. what is secondary data? How did you get to the results in the figure?

This has now been explained in more detailed. The word secondary data has been replaced to explain exactly which data sources were used to get to the results in the figure, as follows:

Firstly, to explore the scope of RSW, data from all sources (qualitative comments in diaries, questionnaires, participant master’s list, RSW case load information and focus group discussion) were used. Any data on patient issues or other actors involved was triangulated, coded and thematically analysed, to identify the variety of issues that RSWs concern themselves with. Themes represented categories of patient issues, and were developed through an iterative, inductive process of comparison of different issues that were coded under the themes. The final coding framework, consisting of 8 domains, is presented as a figure representing the social worker role in the results section.

Results: It is a bit difficult to follow the results. A clearer structure is needed in my opinion. Each method could perhaps be presented: Results from the diary, result from the focus group intervenes/discussion with quotations supporting each theme etc. One overarching theme with subthemes and some explanatory text?

The methodology section has been restructured and explained in more detail. The results follow this same structure, but it is hope that the more detailed explanation of the methodology section makes this clearer. Headings have also been changed to make it clearer which method was used to present the results. To improve the flow, one paragraph about benefits has been split and moved partially under activities (to explain that benefits is not part of the RSW training) and the rest was moved to the section about statutory social work. In addition, findings of the focus group discussion have now been presented as in two large overarching themes (reasons for variation in activity and RSW versus statutory social work) with subthemes and explanatory text underneath those, instead of the smaller headings and paragraphs that were used in the previous draft.

Discussion: page 22 479: The discussion is long and the take home message is quite clear could the discussion therefor perhaps be shortened slightly?

Please clarify regarding the the NICE guidelines. A lot of abbreviations in the text. Perhaps the pre-RRT could be spelled out?

Certain sections of the discussion that were somewhat repetitive have now been removed to shorten the discussion. The NICE acronym has been written in full and the word ‘guidelines’ has been changed to explain clearer that these are quality standards for renal replacement therapy. Pre-RRT has also been written out in full.

The abstract is very clear and describes the results and conclusion really well. The conclusion in the main text lacks the important effect with the renal social worker in the team for the renal patients’ psychological wellbeing.

The following sentence has now been added to the conclusion: RSWs play an important role in reducing distress and supporting the wellbeing of CKD patients.

---

## [Decision Letter · Decision Letter 1]

7 Sep 2022

PONE-D-22-08494R1Exploring the role of the UK renal social worker: the nexus between health and social care for renal patientsPLOS ONE

Dear Dr. Seekles,

Thank you for submitting your manuscript to PLOS ONE. After careful consideration, we feel that it has merit but does not fully meet PLOS ONE’s publication criteria as it currently stands. Therefore, we invite you to submit a revised version of the manuscript that addresses the points raised during the review process.

One reviewer has a further minor comment we ask you to address before we can proceed with publication; please see the reviewers' comments below.

We look forward to receiving your revised manuscript.

Kind regards,

Hugh Cowley

Staff Editor

PLOS ONE

Journal Requirements:

Reviewers' comments:

Reviewer's Responses to Questions

**Comments to the Author**

1. If the authors have adequately addressed your comments raised in a previous round of review and you feel that this manuscript is now acceptable for publication, you may indicate that here to bypass the “Comments to the Author” section, enter your conflict of interest statement in the “Confidential to Editor” section, and submit your "Accept" recommendation.

Reviewer #1: All comments have been addressed

Reviewer #2: All comments have been addressed

2. Is the manuscript technically sound, and do the data support the conclusions?

Reviewer #1: Yes

Reviewer #2: Yes

3. Has the statistical analysis been performed appropriately and rigorously? 

Reviewer #1: Yes

Reviewer #2: Yes

4. Have the authors made all data underlying the findings in their manuscript fully available?

Reviewer #1: Yes

Reviewer #2: Yes

5. Is the manuscript presented in an intelligible fashion and written in standard English?

Reviewer #1: Yes

Reviewer #2: Yes

6. Review Comments to the Author

Reviewer #1: Thank you for responding thoroughly to each comment in my previous review. I still have a minor suggestions to make. Secondary data is mentioned under Methods page 5 but explained on page 8 line 168, I suggest it is explained under Methods. Secondary data: This is also written in the abstract without any explanation, perhaps it should be explained there or deleted?

Reviewer #2: The authors have improved their manuscript taking into account reviewers´ comments and suggestions.

7. PLOS authors have the option to publish the peer review history of their article (what does this mean?). If published, this will include your full peer review and any attached files.

Reviewer #1: No

Reviewer #2: No

---

## [Author Response · Author response to Decision Letter 1]

7 Sep 2022

The listed references have remained the same, no changes have been made.

Reviewer comments:

Thank you for responding thoroughly to each comment in my previous review. I still have a minor suggestions to make. Secondary data is mentioned under Methods page 5 but explained on page 8 line 168, I suggest it is explained under Methods. Secondary data: This is also written in the abstract without any explanation, perhaps it should be explained there or deleted?

We thank the reviewer for their second, thorough assessment of our manuscript. We hope that these changes satisfactorily address the minor comments provided.

The mention of secondary data has been removed on page 5, so now the sentence only explains the difference in the aims of study part 1 and study part 2, as follows:

The first part involved an exploration of the adult RSW role; the second part consisted of an uncontrolled, pre-post evaluation of RSW involvement.

Secondary data are explained in detail under the section methods and data collection tools. 

In the abstract, an explanation has been added to clarify that secondary data includes data on caseloads and audit files where available.

---

## [Editor Report · Decision Letter 2]

9 Sep 2022

Exploring the role of the UK renal social worker: the nexus between health and social care for renal patients

PONE-D-22-08494R2

Dear Dr. Seekles,

We’re pleased to inform you that your manuscript has been judged scientifically suitable for publication and will be formally accepted for publication once it meets all outstanding technical requirements.

Kind regards,

Hugh Cowley

Staff Editor

PLOS ONE
---

## [Editor Report · Acceptance letter]

15 Sep 2022

PONE-D-22-08494R2 

Exploring the role of the UK renal social worker: the nexus between health and social care for renal patients 

Dear Dr. Seekles:

I'm pleased to inform you that your manuscript has been deemed suitable for publication in PLOS ONE. Congratulations! Your manuscript is now with our production department. 

Kind regards, 

on behalf of

Mr Hugh Cowley 

Staff Editor

PLOS ONE